# Comparative Analysis of the Antioxidant, Antidiabetic, Antibacterial, Cytoprotective Potential and Metabolite Profile of Two Endophytic *Penicillium* spp.

**DOI:** 10.3390/antiox12020248

**Published:** 2023-01-22

**Authors:** Kumar Vishven Naveen, Kandasamy Saravanakumar, Anbazhagan Sathiyaseelan, Myeong-Hyeon Wang

**Affiliations:** Department of Bio-Health Convergence, Kangwon National University, Chuncheon 24341, Republic of Korea

**Keywords:** endophytes, *Penicillium* sp., oxidative stress, bacterial infection, diabetes

## Abstract

The current study assessed the metabolite abundance, alpha (α)-amylase and α-glucosidase inhibitory, antioxidant, and antibacterial activity of the ethyl acetate extract (EAE) of endophytic *Penicillium lanosum* (PL) and *Penicillium radiatolobatum* (PR). A higher extract yield was found in EAE-PR with a total phenolic content of 119.87 ± 3.74 mg of GAE/g DW and a total flavonoid content of 16.26 ± 1.95 mg of QE/g DW. The EAE-PR inhibited α-amylase and scavenged ABTS+ radicals with a half-maximal inhibitory concentration (IC_50_) of 362.5 and 37.5 µg/mL, respectively. Compared with EAE-PL, EAE-PR exhibited higher antibacterial activity against Gram-positive and Gram-negative pathogens. Treatment with EAE-PR (1000 µg/mL) did not cause significant toxicity in the HEK-293 cell line compared to the control cells (*p* < 0.05). EAE-PR treatments (250–1000 µg/mL) showed higher cytoprotective effects toward H_2_O_2_-stressed HEK-293 cells compared with ascorbic acid (AA). The UHPLC-Q-TOF-MS/MS analysis revealed the presence of thiophene A (C_13_H_8_S), limonene (C_10_H_16_), and phenylacetic acid (C_8_H_8_O_2_) in EAE-PR. Furthermore, these compounds demonstrated substantial interactions with diabetes (α-amylase and α-glucosidase), oxidative stress (NADPH-oxidase), and bacteria (D-alanine D-alanine ligase)-related enzymes/proteins evidenced in silico molecular docking analysis.

## 1. Introduction

According to the World Health Organization (WHO), almost 74% of deaths are due to non-communicable diseases (NCDs), including chronic respiratory diseases, cardiovascular diseases, diabetes, and cancer [1]. The NCDs are associated with the free radicals, called reactive oxygen species (ROS) and reactive nitrogen species (RNS), namely hydroxyl (HO˙), alkoxyl (RO˙), peroxyl (ROO˙), superoxide (O_2_ˉ), nitric oxide (NO˙), and hydrogen peroxide (H_2_O_2_), generated by numerous physiological progressions in the body [2]. The ROS and RNS cause unavoidable damage to cellular components (DNA, proteins, enzymes, lipids, carbohydrates, etc.) and living tissues via oxidative damage or oxidative stress, which implicates various inflammatory and degenerative diseases in the biological system, such as mellitus, Alzheimer’s disease, Parkinson’s disease, atherosclerosis, liver cirrhosis, and depression [2,3]. Free radicals are exceptionally unstable and highly reactive molecules as they have unpaired valence electrons in their atoms’ outermost shell and are momentary and interact non-selectively with their neighboring molecules [4,5]. Antioxidants are agents that scavenge free radicals (oxidants) and/or transform ROS/RNS initiators (e.g., hydroperoxides) into non-reactive products. Although some antioxidants are naturally produced in the human body (endogenous), in diseased/deficient conditions antioxidants are required through the diet or supplementation (exogenous) to maintain/establish the redox balance. Therefore, natural antioxidants are preferred over synthetic ones and have attracted researchers’ attention.

Diabetes is one of the main chronic diseases among the NCDs, affecting nearly 537 million adults worldwide. The International Diabetes Federation (IDF) and WHO estimated that the global diabetes prevalence would exceed 783 million people by 2045 [1,6]. Diabetes is characterized by constant hyperglycemia that may result from an immune system attack (autoimmune disease) on pancreatic beta-cells (Type-1) or may be associated with sugar accumulation in the blood and insulin resistance in the body (Type-2) [7]. However, chronic hyperglycemia induces auto-oxidation in glucose and generates free radicals, resulting in microvascular/macrovascular complications and/or apoptosis of endothelial cells, and may induce atherosclerosis, cardiovascular complications, hypertension, retinopathy, and renal disorders [8]. Type-2 diabetes can be controlled by changing lifestyle and dietary habits to balance insulin secretion and blood glucose levels. In addition, inhibiting the activities of certain enzymes that convert dietary polysaccharides and monosaccharides into simple sugars (α-amylase and α-glucosidase) has shown promise in diabetes treatment [9]. For instance, α-amylase and α-glucosidase inhibitors, namely acarbose, miglitol, and voglibose, are used in current diabetic medications; however, they produce gastrointestinal disturbances in hosts [7]. Therefore, bioprospecting of novel α-amylase and α-glucosidase inhibitors with minimal side effects could be vital for type-2 diabetes management. Recently, in the search for new antidiabetic agents, the outcomes of natural product research have been encouraging [7,8,9,10,11].

The discovery and development of therapeutic agents from natural sources have historically relied on plants. However, the authentication, cultivation, and procurement of plant-based compounds are costly, laborious, and time-consuming processes and involve ecological concerns. However, plants accommodate certain microbes (actinomycetes, bacteria, and fungi) asymptomatically within the tissues or intracellular spaces known as endophytes. Endophytic fungi are considered to be polyphyletic groups of microbes that carry a common biosynthetic pathway of secondary metabolites similar to that of plants via stabilizing a mutualistic relationship through metabolic interchange and unique plant–microbe interactions [12]. Among the endophytic fungi, the *Penicillium* spp. are known to produce antagonist compounds that improve plants’ immunity and the resultant protection against abiotic stress and numerous diseases [13]. Moreover, the secondary metabolites from *Penicillium* sp. have a high degree of structural diversity, including alkaloids, fatty acids, macrolides, peptides, phenylpropanoids, polysaccharides, polyketides, terpenoids, and other structure classes [14]. In addition, endophytic *Penicillium* sp.-based compounds are reported to have distinct biological activities and therapeutic applications [15]. In addition, fungal endophytes require a nominal amount of labor and culture media, provide the straightforward isolation of culture-dependent metabolites and simple production scale-up, and are non-toxic to mammalian systems.

In addition to NCDs, common bacterial infections are causing a high mortality rate. In 2019, bacterial infections were associated with one in eight fatalities worldwide [16]. According to the Centers for Disease Control and Prevention (CDC), 48 million individuals are infected by foodborne infections each year in the United States [17]. *Escherichia coli*, *Salmonella* sp., *Staphylococcus aureus*, and *Listeria monocytogenes* are among the major foodborne pathogens and threaten public health [18]. In that respect, polyphenolics such as phenolic acids exhibit an antibacterial action via alteration of the cell membrane potential through hyperacidification at the plasma membrane interphase and sodium–potassium ATPase pump alteration through intracellular acidification against pathogens [19]. Recent studies have also indicated that polyphenolics derived from endophytic *Penicillium* sp. have antibacterial effects on Gram-positive and Gram-negative pathogens [20,21,22]. Additionally, some structurally diverse and bioactive metabolites, such as epoxydon, griseofulvin, javanicin, meleagrin, oxyskyrin, and penicillic acid are found in extracts of endophytic *Penicillium* sp. [15]. Moreover, Nischitha et al. detected bilirubin, diisobutylphthalate, and hexadecanamide in the organic extract of *P. pinophilum* and reported their antibacterial activity [22].

Considering all these factors, this study aimed to evaluate the antidiabetic, antioxidant, and antibacterial potential of the endophytic *Penicillium* sp. extract. Thus, the secondary metabolites from two endophytic *Penicillium* strains (*P. lanosum* strain AN001 and *P. radiatolobatum* strain AN003) were extracted, tested for bioactivity, and compared. Additionally, the compounds in the extracts were elucidated using UHPLC-Q-TOF-MS/MS, and we studied the in silico molecular interactions of bioactive compounds with enzymes/proteins associated with diabetes, oxidative stress, and bacteria.

## 2. Materials and Methods

### 2.1. Reagents and Consumables

Mueller–Hinton agar (MHA) and Mueller–Hinton broth (MHB) were supplied by KisanBio Co., Ltd. (MB cell), Seoul, Republic of Korea (ROK). Peptone (BactoTM) and potato dextrose agar (PDA; DifcoTM) were obtained from Becton, Dickinson and Company (Franklin Lakes, NJ, USA). Ethyl acetate (EA), ethyl alcohol (EtOH), and L (+)-ascorbic acid (AA) were purchased from Daejung Co., Ltd. (Busan, ROK). Dimethyl sulfoxide (DMSO) was obtained from Duchefa Biochemie (Haarlem, The Netherlands). Dextrose, acarbose (AC), erythromycin (Ery), starch, α-amylase from porcine pancreas, α-glucosidase from *Saccharomyces cerevisiae*, 3,5-dinitrosalicylic acid (DNS), *p*-nitrophenyl-α-D-glucopyranoside (PNPG), sodium carbonate (Na_2_CO_3_), aluminum chloride (AlCl_3_), Folin–Ciocalteu’s phenol reagent, potassium acetate (CH_3_COOK), 2,2-diphenyl-1-picrylhydrazyl (DPPH), 2,2′-azino-bis(3-ethylbenzothiazoline-6-sulfonic acid) diammonium salt (ABTS), 2,2′-azo-bis(2-amidino propane dihydrochloride (ABAP), thiazolyl blue tetrazolium bromide (MTT), propidium iodide (PI), acridine orange (AO), ethidium bromide (EB), rhodamine-123 (Rh-123), and dichlorofluorescein-diacetate (DCFH-DA) were supplied by Sigma-Aldrich (Yongin, ROK). Fetal bovine serum (FBS) and penicillin–streptomycin (PS) solution were purchased from Hyclone Laboratories (Logan, UT, USA). Phosphate-Buffered Saline (PBS) was obtained from Corning^®^ (Manassas, VA, USA). Dulbecco’s Modified Eagle Medium (DMEM) and Roswell Park Memorial Institute (RPMI) medium were acquired from ThermoFisher Scientific (Seoul, ROK). The human embryonic kidney 293 (HEK-293) cells were procured from the Korean Cell Line Bank (KCLB, Seoul, ROK). Pathogenic bacterial strains, namely *Bacillus cereus* (ATCC 14579), *Staphylococcus aureus* (ATCC 19095), *Listeria monocytogenes* (ATCC 15313), *Escherichia coli* (ATCC 43888), and *Salmonella enterica* (ATCC 14028), were supplied by the Korean Culture Center of Microorganisms (KCCM, Seoul, ROK).

### 2.2. Fungal Culture Medium, Liquid Fermentation, and Metabolites Extraction

The potato dextrose broth (PDB) used for fungal cultivation was prepared according to the earlier report [23] with minor modification. Briefly, the potato infusion was prepared by boiling the sliced potatoes (200 g/L) for 30 min and filtering them using cotton gauze. The potato infusion, dextrose (10 g/L), and peptone (2 g/L) were mixed, the pH was adjusted to 5.1, and the solution was autoclaved at 121°C for 15 min. The formerly isolated endophytic isolates, namely *P. lanosum* strain AN001 (GenBank accession ID. MW237700) and *P. radiatolobatum* strain AN003 (GenBank accession ID. MW237703), were cultured on freshly prepared PDA plates for 4 days at 25 ± 2 °C. From the plates of both fungi, mycelial edges were cut (12 plugs, each with a diameter of 15 mm) and separately inoculated in Erlenmeyer flasks containing PDB (3 L for each fungus). The culture flasks were transferred to a shaker incubator (140 rpm and 25 ± 2 °C) for 3 days and then incubated statically for 14 days at 25 ± 2 °C. After incubation, the mycelia were separated from the culture broth using cotton gauze and then the culture broth was filtered through Whatman filter paper No.1 to obtain the mycelium-free filtrate. Fungal metabolites were extracted three times using EA solvent as described elsewhere with minor modifications [24]. In brief, the filtrate and EA were added (*v*/*v*) and agitated for 5–6 min in a separatory funnel, and then the upper (organic) layers were collected by separating the lower (aqueous) layers. The organic layers (EA extract) of both fungi were merged separately and condensed by evaporating the EA through a rotary evaporator at 38 °C. Finally, the EA extract of *P. lanosum* (EAE-PL) and the EA extract of *P. radiatolobatum* (EAE-PR) were collected, weighed, and kept at 4 °C for further experimentation.

### 2.3. Assessment of Total Phenolics and Flavonoid Content

The total phenolic content (TPC) and total flavonoid content (TFC) in the EAE-PL and the EAE-PR were analyzed according to a previous report [24] with slight modifications as described in the Appendix A.

### 2.4. α-amylase and α-glucosidase Inhibition Assay

The α-amylase and α-glucosidase inhibitory activity of the EAE-PL and EAE-PR were evaluated according to a previously described protocol [8] as described in the Appendix A.

### 2.5. Antioxidant Assays

#### 2.5.1. ABTS+ and DPPH Radical Scavenging Assay

The ABTS+ and DPPH radical scavenging activity of the EAE-PL and EAE-PR were evaluated as described earlier [8] and in the Appendix A.

#### 2.5.2. Ferric-Reducing Power Assay

The ferric ion reducing potential of the EAE-PL and EAE-PR was assessed as reported previously [24,25]. In short, 500 µL of K_3_[Fe(CN)_6_] solution (30 mM) was mixed with 175 µL of PBS (0.6 M, pH 6.6), added to 100 µL of sample (EAE-PL and EAE-PR) separately, and then incubated for 20 min at 50 °C. Following incubation, 500 µL of freshly prepared trichloroacetic acid (10%) was added to the reaction mix and mixed well. Then, added 1500 µL of FeCl_3_ (0.1%) and mixed the solution. Finally, the absorbance was recorded at 700 nm. AA was taken as a positive control.

#### 2.5.3. Determination of Peroxyl Radical Scavenging Capacity (PSC)

The peroxyl radical scavenging capacities of the EAE-PL, EAE-PR, and AA were assessed as reported previously [26] and compared. In short, 100 µL (1 mg/mL) of sample (EAE-PL, EAE-PR, and AA) was thoroughly mixed with 100 µL of DCFH-DA (2.48 mM) separately. The reaction was started by adding 50 µL of ABAP (200 mM) at 37 °C. The fluorescence of the reaction mix was read every 2 min at 538 nm after excitation at 485 nm for 40 min using a SpectraMax i3 microplate reader (Molecular Devices, Seoul, Republic of Korea).

### 2.6. Antibacterial Susceptibility Test

The antibacterial susceptibility of the EAE-PL and EAE-PR against Gram-positive (*B. cereus*, *S. aureus*, and *L. monocytogenes*) and Gram-negative (*E. coli* and *S. enterica*) pathogens was detected by a disc diffusion assay as per the Clinical Laboratory Standard Institute (CLSI) guidelines. Bacterial strains were cultured in NB medium and grown overnight at 37 ± 2 °C. The next day, 100 µL of bacterial suspension (1.5 × 10^4^ CFU/mL) was spread onto fresh MHA plates using a sterile glass spreader. Then, 50 µL (500–1000 µg/mL) of test sample (EAE-PL and EAE-PR) was loaded onto sterile paper discs (6 mm diameter) and air dried. The sample-containing paper discs were placed on the bacterial plates and incubated at 37 ± 2 °C for 18 h. The ery (100 µg/mL) was taken as an positive control. PBS (50 µL) treatment was taken as an experimental negative control. Following incubation, the zone of growth inhibition was measured (in mm) around the discs using a Vernier scale.

#### Assessment of Minimum Inhibitory Concentration (MIC) and Minimum Bactericidal Concentration (MBC)

The MIC and MBC of the EAE-PL and EAE-PR were determined on pathogenic strains (*B. cereus*, *S. aureus*, *L. monocytogenes*, *E. coli*, and *S. enterica*) using a microdilution assay. Each well of a 96-well flat bottom plate was filled with 100 µL of freshly prepared MHB. A bacterial suspension (1.5 × 10^4^ CFU/mL) that had been grown overnight was added (10 µL) to the respective wells and treated (10 µL) with varying concentrations (0–1000 µg/mL) of test sample (EAE-PL and EAE-PR). The Ery (100 µg/mL) treated (10 µL) wells were taken as an experimental positive control and wells containing the bacterial suspension with PBS (10 µL) were considered to be a negative control. Then, the plates were incubated at 37 ± 2 °C, and the absorbance (at 600 nm) was recorded using a UV-vis microplate reader over time (0–24 h). Afterward, the MBC of the EAE-PL and EAE-PR was ascertained on the basis of turbidity and bacterial survival. From the wells without bacterial growth (no visible turbidity), 50 µL of suspension was streaked onto fresh MHA plates and maintained for 24 h at 37 ± 2 °C. Later, the bacterial colony growth was observed on the plates to determine the MBC value.

### 2.7. Cell Viability Analysis (MTT Assay)

The effect of EAE-PL, EAE-PR, and AA on the viability of HEK-293 cells was tested in an MTT assay as reported elsewhere [27]. Briefly, the cells were cultured in DMEM supplemented with FBS (10%) and PS (1%) antibiotics and incubated at 37 ± 1 °C in a humidified CO_2_ (5%) atmosphere. Exponentially growing cells (70–80% confluence) were seeded (10^4^ cells/100 µL) in a 96-well plate (flat-bottom) and grown overnight under the above-mentioned conditions. The next day, the cells were separately treated (10 µL/well) with the serially diluted concentration (7.8–1000 µg/mL) of EAE-PL, EAE-PR, and AA and placed in the incubator for 12 h. Untreated cells were considered to be an experimental control. Following incubation, the MTT solution (5 mg/mL) was added (10 µL) to each well in the dark and incubated for 3 h. Later, the medium was removed from the wells, and 100 µL of DMSO was added to dissolve the Formazan crystals formed by the cells. The absorbance was recorded at a wavelength of 570 nm using a Microplate reader (SpectraMax^®^ ABS Plus, Molecular Devices, San Jose, CA, USA) and the cell viability (%) was calculated.

#### Cell Viability Analysis (Fluorescent Staining Assay)

A fluorescent staining examination was used to investigate the physiological impact of the EAE-PL, EAE-PR, and AA on HEK-293 cells as stated elsewhere [28]. In brief, the cells were grown on a 24-well culture plate (4 × 10^4^ cells/500 µL media/well) overnight at 37 ± 1 °C in a humidified CO_2_ (5%) incubator. The wells were treated (50 µL) with EAE-PL, EAE-PR, and AA at a concentration of 125 µg/mL and further incubated for 18 h. The nuclear variation, apoptotic stages, and necrotic damage were examined using AO/EB dual staining. Rh-123 staining was used to observe the alteration in the mitochondrial membrane potential, and DCFH-DA staining was used to analyze the ROS status. Images were captured on an Olympus CKX53 culture microscope (Tokyo, Japan).

### 2.8. Cytoprotective Activity against H_2_O_2_ Stress

The cytoprotective effect of the EAE-PL and EAE-PR against H_2_O_2_ induced stress was assessed in HEK-293 cells using an MTT assay. The cells were cultivated, seeded in a 96-well plate (flat-bottom), grown, treated, and experimented upon as previously mentioned (in the Cell Viability Analysis (MTT assay) section). Cells were treated (10 µL/well) with H_2_O_2_ (1.6 mM) and incubated for 30 min. Then, we added EAE-PL, EAE-PR, and AA (10 µL/well) at a serially diluted concentration (7.8–1000 µg/mL) and incubated the cells for 12 h. Finally, the cell viability (%) was calculated. The AA-containing wells were considered to be a positive control and only the H_2_O_2_-treated cells were considered to be a negative control. Later, the cytoprotective activity of the samples (125 µg/mL) against H_2_O_2_-stressed HEK-293 cells was substantiated by light and fluorescent staining examinations as described in the cell viability analysis (fluorescent staining assay) section. The cellular morphology was visualized using a light microscope, and fluorescent stains, namely PI and DCFH-DA, were used to distinguish dead cells and analyze the ROS status, respectively.

### 2.9. Metabolite Profiling by UHPLC-Q-TOF-MS/MS

Bioactive compounds of the EAE-PL and EAE-PR were analyzed through an ultra-high-performance liquid chromatograph equipped with a quadrupole time-of-flight mass spectrometer (UHPLC-Q-TOF-MS/MS) using a SCIEX ExionLCTM AD series-equipped X500R Q-TOF system (Framingham, MA, USA) [29]. The samples (EAE-PL and EAE-PR) were prepared at a concentration of 1000 μg/mL by dissolution in EtOH (70%) and then filtered using syringe filters (0.2 µm). The samples (10 µL) were injected into the autosampler (100 × 3 mm, Accucore^TM^ C-18 column) and eluted through binary mobile phases A (water and 0.1% formic acid) and B (methanol) at a flow rate of 0.4 mL/min in a linear gradient program (25 min). The MS analysis was performed in both electrospray ionization modes (ESI+ and ESI-). In the (+) ionization mode, the voltage (+40 V) and capillary temperature (320 °C) were set with a spray voltage of 3.6 kV. In the (−) ionization mode, the voltage, capillary temperature, and spray voltage were set to −40 V, 320 °C, and 2.7 kV, respectively. Spectral libraries such as Metlin (XCMS Online), Pubchem, and ChemSpider were utilized for the identification of compounds.

### 2.10. Molecular Docking Analysis

The possible mechanism behind the bioactivities of the tentatively identified fungal compounds was assessed through an in silico molecular docking study. Briefly, to prepare the structures of the selected compounds (ligands), the colonial smiles were retrieved from NIH PubChem. Then, using UCSF chimera software (1.16), the structure of the ligands was designed, the energy was minimized, and the structures were saved as mol files. The crystal structure (protein data bank (PDB) files) of target proteins, such as α-amylase (PDB: 1OSE), α-glucosidase (PDB: 5NN8), NADPH-oxidase (PDB: 2CDU), and D-alanine D-alanine ligase (PDB: 2PVP), were downloaded from the RCSB PDB database. The molecular interactions among the ligands (mol files) and target proteins (PDB files) were carried out in the ArgusLab program (V.4.0.1). The water molecules and miscellaneous residues were removed from the target proteins and the binding packet was set before the docking run. Docking results were visualized using the BIOVIA Discovery Studio Visualizer (V.21.1.0.20298) tool. Acarbose (C_25_H_43_NO_18_), ascorbic acid (C_12_H_16_O_13_), and erythronolide A (C_21_H_38_O_8_) were used as positive controls.

### 2.11. Statistical Analysis

Each experiment was run in triplicate, and the results are shown as the mean ± standard deviation. The samples’ significance was calculated using the one-way analysis of variance (ANOVA; *p* < 0.05). The ‘prcomp’ program was used to perform Principle Component Analysis (PCA) to explain the variance in the data set. Using Ward’s linkage and Pearson’s correlation program, hierarchical cluster analysis was performed to produce heatmaps. The analysis of the metabolite pathway was performed based on the *p*-value (0~1) using the Kyoto Encyclopedia of Genes and Genomes (KEGG) compound database annotation in the OmicStudio web-tool.

## 3. Results and Discussion

Endophytic fungi are among the most important components in the plant–microbe interaction ecosystem and are also rich sources of natural products with pharmacological properties [20]. Secondary metabolites are of great importance due to their anticancer, antioxidant, and antibacterial activities. From an endophytic fungus fermentation broth, secondary metabolites with different compounds, contents, and quantities can be isolated using various solvents. The polar and medium-polar solvents are more effective at extracting low- and high-molecular-weight polyphenols compared with non-polar solvents [10]. However, EA is reported to be the most effective solvent for extracting mid-polar secondary metabolites with a higher yield and polyphenol content [8,11,20].

### 3.1. Total Yield and Polyphenolics Quantification

An analysis of the total yield, total phenolic content, and total flavonoid content of the EAE-PL and EAE-PR is presented in Table 1. The extract yield was noted to be 0.41 ± 0.08% and 0.63 ± 0.05% for the EAE-PL and EAE-PR, respectively. Polyphenols contain hydroxyl groups with a hydrogen-donating capacity and possibly contribute to antioxidation, free radical scavenging, and stabilizing lipid oxidation [30,31]. The TPC of the EAE-PL was determined to be 17.89 ± 1.63 mg of gallic acid equivalents, dry weight (GAE/g DW) and the TPC of the EAE-PR was noted to be 119.87 ± 3.74 mg of GAE/g DW (Table 1). The TFC of the EAE-PL and EAE-PR was determined to be 3.31 ± 0.96 mg of quercetin equivalents, dry weight (QE/g DW) and 16.26 ± 1.95 mg of QE/g DW, respectively (Table 1). It is evident that flavonoids are one of the major compounds in the overall polyphenol composition of natural extracts and are associated with antioxidant, antibacterial, anti-inflammatory, and immunoregulatory activities [32].

### 3.2. α-amylase and α-glucosidase Inhibition Activities

α-amylase and α-glucosidase are hydrolytic enzymes known to catalyze the hydrolysis of starch and disaccharides, respectively. In humans, α-amylases hydrolyze starch into maltodextrin and glucose, while α-glucosidases hydrolyze disaccharides (sucrose and maltose) into monosaccharides (fructose and glucose). Therefore, inhibiting the activity of α-amylase and α-glucosidase controls the rise in blood sugar levels and prevents postprandial hyperglycemia and type-2 diabetes [10]. However, fungal compounds such as alkaloids, flavonoids, fatty acids, phenolic acids, and triterpenes have been reported to be potential α-amylase and α-glucosidase inhibitors [7]. This study evaluated the α-amylase and α-glucosidase inhibitory activities of the EAE-PL and EAE-PR compared to AC as presented in Appendix A. The inhibition (%) activities of the tested samples were found to be proportional to the tested concentration for both α-amylase (Appendix A) and α-glucosidase (Appendix A) enzymes. Yet, the half-maximal inhibitory concentration (IC_50_) of the EAE-PL for α-amylase and α-glucosidase inhibition could not be determined up to the highest tested concentration (1000 µg/mL) (Table 2). However, the EAE-PR inhibited the activity of α-amylase and α-glucosidase with an IC_50_ value of 362.5 and 525 µg/mL, respectively. Meanwhile, AC exhibited a lower IC_50_ value for α-amylase (37.5 µg/mL) and α-glucosidase (87.5 µg/mL) inhibition (Table 2). An earlier study also reported the α-glucosidase inhibitory activity of the EA of endophytic P. citrinum to have an IC_50_ value of 2800 µg/mL [33].

### 3.3. Antioxidant Activities

Antioxidants help to maintain regular metabolism in the mammalian system by preventing oxidative stress and associated ailments. In particular, antioxidants act as the substrate for the incomplete oxygen reduction of ROS production in mitochondria and prevent oxidative damage to cells [24,34]. However, a single antioxidant experiment remains insufficient to comprehensively anticipate the antioxidant efficacy of natural extracts. So, the antioxidant potentials of the EAE-PL and EAE-PR were evaluated through ABTS+ and DPPH radical scavenging, ferric-reducing power, and PSC assays as can be seen in Appendix A and Table 2.

The DPPH and ABTS+ assays are traditional, rapid, and renowned methods for determining antioxidant activity. The antioxidants function as electron donors to neutralize the DPPH free radicals and scavenge the ABTS+ radicals by inhibiting the activity of potassium persulphate [22]. Experimental findings indicate the concentration-dependent ABTS+ and DPPH radical scavenging activity of the tested samples (EAE-PL, EAE-PR, and AA) (Appendix A). The IC_50_ of the EAE-PL and EAE-PR for ABTS+ radical scavenging was determined to be 59.5 and 37.5 µg/mL, respectively, compared with AA (49.25 µg/mL) (Table 2). However, for the EAE-PL, the IC_50_ value for DPPH radical scavenging could not be determined up to 1000 µg/mL, yet the EAE-PR exhibited an IC_50_ value of 187.5 µg/mL, compared with AA (241.6 µg/mL) (Table 2). It should be noted that ABTS and DPPH radicals are structurally different, and some organic compounds, such as dihydrochalcones and flavanones, do not react with DPPH radicals, unlike ABTS radicals [35]. This could be the possible reason for the significant differences in the ABTS+ and DPPH radical scavenging activities (IC_50_) of the EAE-PL and EAE-PR.

The reducing powers of the antioxidants are attributed to their electron donation capability and free radical quenching capacity [36]. The ferric-reducing potential of the tested samples (EAE-PL, EAE-PR, and AA) was also found to be concentration-dependent (Appendix A) with an IC_50_ value of 733.3, 175.0, and 326.5 µg/mL for the EAE-PL, EAE-PR, and AA, respectively (Table 2). Similarly, a previous study reported the ferric reducing potential of the EAE of endophytic *Paraconiothyrium brasiliense* to have an IC_50_ value of 358 ± 1.56 µg/mL [24]. It is believed that, in organic extracts, phenolic acids that contain aromatic rings with carboxyl and hydroxyl residues hold ROS-neutralizing and transition-metal-reducing powers. However, nonmethylated phenolic acid derivatives are reported to have higher ferric-reducing powers compared with methylated ones [37].

The PSC assay is a simple, rapid, and sensitive technique by which to evaluate the ability of organic extracts to scavenge peroxyl radicals and is considered superior to DPPH, ABTS, and FRAP assays because it represents naturally occurring ROS and physiological parameters (pH and temperature) of mammalian systems [26]. The peroxyl radical scavenging activity of the tested samples (EAE-PL, EAE-PR, and AA) was observed to be time-dependent (up to 40 min) (Appendix A). The EAE-PR and AA exhibited over 90% of the PSC when observed at the 40th minute (Appendix A). The PSC (IC_50_) of the EAE-PR (532.36 µg/mL) was found to be nearly equal to that of the AA (521.86 µg/mL) and substantially higher than that of the EAE-PL (>1000 µg/mL) (Table 2). The natural extracts contain a mixture of substances, carbohydrates, and peptide contents that might affect their antioxidant activities via a potential synergistic action compared with individual substances [35]. Additionally, our results suggest a significant correlation between total polyphenolic content and antioxidant potency and corroborate previous findings [8,10,11,20,31].

### 3.4. Antibacterial Activity

The disc diffusion assay is a qualitative approach to defining the susceptibility of bacterial strains to an antibacterial agent. In the past, natural products played a significant role in the discovery and development of antibacterial agents. The rapid growth of resistant bacterial strains has required a re-evaluation of these systems as a means of identifying novel chemical skeletons with antibacterial activity for the development of new drugs [38]. The bacterial susceptibility to the EAE-PL and EAE-PL was measured as the zone of inhibition (ZoI) (Figure 1 and Table 3). All the tested strains exhibited a ZoI towards the highest tested concentration (1000 µg/mL) of the EAE-PL and EAE-PR (Figure 1a–e). Although the treatment with 500 µg/mL of EAE-PL showed a ZoI against *B. cereus* (9 ± 0.5 mm), a significant ZoI could not be detected against *S. aureus* (0 mm), *L. monocytogenes* (0 mm), *E. coli* (6.5 ± 0.2 mm), or *S. enterica* (0 mm). However, the ZoI ranged from 8 ± 0.3 to 11.5 ± 1 mm against the tested pathogens for the EAE-PR (500 µg/mL) treatment. However, Ery (100 µg/mL) exhibited a ZoI range (21 ± 0.8–22 ± 1.5 mm) against the tested pathogens (Table 3). Phenolic compounds from natural sources possess a wide range of structural characteristics, including the presence/absence, substitution position, and number of hydroxyl groups as well as the length of saturated side chains, all of which possibly contribute to their antibacterial action [39]. The findings of the disc diffusion assay suggest that EAE-PL and EAE-PR have bacterial-strain-specific and concentration-dependent (500–1000 µg/mL) antibacterial activity.

The microtiter assay is a quantitative approach to defining the MIC and MBC of bacterial strains to an antimicrobial agent. The MIC and MBC of the EAE-PL and EAE-PR were determined as can be seen in Appendix A and Table 4. The MIC (µg/mL) of the EAE-PL was determined for *B.cereus*, *S. aureus*, *L. monocytogenes*, *E. coli*, and *S. enterica* to be 62.5, 500, 500, 125, and 250, respectively. The EAE-PR exhibited comparatively lower MIC values, such as 31.25 µg/mL (*B. cereus*), 125 µg/mL (*S. aureus*), 31.25 µg/mL (*L. monocytogenes*), 62.5 µg/mL (*E. coli*), and 62.5 µg/mL (*S. enterica*) (Appendix A and Table 4). Similarly, in a recent report, the EA extract of the endophytic *P. oxalicum* exhibited an MIC range of 500–2000 µg/mL against *E. coli*, *Pseudomonas aeruginosa*, *S. aureus*, and *B. subtilis* [20]. Furthermore, the EAE-PL exhibited higher MBC values compared with the EAE-PR. For the EAE-PL, only *B. cereus* and *E. coli* exhibited an MBC of 500 µg/mL, whereas the MBC against *S. aureus*, *L. monocytogenes*, and *S. enterica* could not be determined below 1000 µg/mL. Moreover, the MBC of the EAE-PR was determined to be 125 µg/mL, 500 µg/mL, 125 µg/mL, 250 µg/mL, and 250 µg/mL for *B. cereus*, *S. aureus*, *L. monocytogenes*, *E. coli*, and *S. enterica*, respectively (Appendix A and Table 4). Inhibition of the apparent growth (colony) of bacterial cells may be due to a disruption of cell membrane integrity or suppression of DNA and/or protein synthesis resulting in cell death [21,22]. It should be noted that the genetic makeup and phenotypic traits of bacterial cells determine their response to different antimicrobial agents [40]. Moreover, the difference in chemical composition could also be the reason for the different antibacterial effects of the EAE-PL and EAE-PR against the tested bacterial cells (Figure 1, Appendix A and Table 3 and Table 4).

### 3.5. Cell Viability Analysis

Endophytic fungi are known to produce strong cytotoxic metabolites that raise concerns about the therapeutic application of their extracts. Hence, the effect of the EAE-PL, EAE-PR, and AA on the viability of HEK-293 cells was tested and is presented in Figure 2. The outcome of the MTT assay indicated a trivial reduction in the viability of HEK-293 cells at the tested concentrations (250–1000 µg/mL) of samples (EAE-PL, EAE-PR, and AA) compared with untreated (control) cells. Although the treatment with AA (1000 µg/mL) resulted in a decrease (~30%) in the viability of cells, the treatment with a lower dose of AA (15.6 µg/mL) and EAE-PL (62.5 µg/mL) increased the viability (>100%) compared with control cells (Figure 2a). The non-toxicity of the EAE-PL, EAE-PR, and AA towards HEK-293 cells could be accredited to their antioxidant behavior. Additionally, natural extracts have the characteristic of selective toxicity towards different cell types. For instance, the EAE of endophytic *Paraconiothyrium brasiliense* is highly toxic to prostate cancer cells (PC3) but non-toxic to HEK-293 cells [24].

Furthermore, the findings of the MTT assay on HEK-293 cells were corroborated by the fluorescent staining assay as can be seen in Figure 2b–d. In the AO/EB dual staining assay, the treatment with EAE-PL, EAE-PR, and AA did not show a difference in cellular morphology compared to the control cells. However, the EAE-PR-treated cells demonstrated an insignificant number of apoptotic cells (orange fluorescence) (Figure 2b). However, the Rh-123 stain did not evidence a noteworthy mitochondrial membrane potential loss in the EAE-PL, EAE-PR, and AA-treated cells compared to the control cells (Figure 2c). In the DCFH-DA staining results, the EAE-PR-treated cells exhibited an inconsequential yet higher ROS level (intense green fluorescence) compared with the other treatments (EAE-PL and AA) and the control cells (Figure 2d).

### 3.6. Cytoprotective Effect against H_2_O_2_ Stress

It is evident that natural extracts can exhibit different antioxidant capacities in distinct chemical and biological systems (in vitro, ex vivo, and in vivo) [41,42]. In that respect, cell-based assays offer a complex and more physiologically and biologically realistic system as the cellular environment can imitate disease states, sustain a signaling cascade, and simulate the tested compounds’ response. Therefore, the cytoprotective activity of the EAE-PL, EAE-PR, and AA was tested in HEK-293 cells subjected to H_2_O_2_ stress (Figure 3). The H_2_O_2_ treatment of HEK-293 cells showed cell viability (~30%) compared with control cells (100%). Dose-dependent protection was observed against the treated (7.8–1000 µg/mL) samples and the cell viability was increased to 43%, 59%, and 55% by 125 µg/mL of EAE-PL, EAE-PR, and AA, respectively, indicating 13%, 29%, and 25% recovery from H_2_O_2_-induced cell damage (Figure 3a). However, at higher concentrations (125–1000 µg/mL), the EAE-PR exhibited a greater cytoprotective effect compared with AA. This is also in line with previous studies suggesting that the combination of AA and H_2_O_2_ might also act as a pro-oxidant [43,44].

In addition, the cytoprotective activity of the EAE-PL, EAE-PR, and AA was substantiated through light microscopy and PI and DCFH-DA staining assays (Figure 3b–d). During light microscopy, the control cells appeared in large numbers with an organized cellular morphology, such as a complete cell structure, a dense cytoplasm, and regular arrangement and attachment. However, H_2_O_2_-treated cells were found in reduced numbers and with an atrophied cell morphology. The EAE-PL-treated cells were also observed to have a changed cell structure and severe damage. However, the EAE-PR and AA-treated cells were found to have an improved cell density and recovered from the abnormal cell structure (Figure 3b). Using the nucleic acid stain PI, dead cells can be distinguished (red fluorescence) in a population of cells [10]. Compared with the control cells, severe cell death (a higher number of cells with a red fluorescence) was observed in the H_2_O_2_ treatment group. Moreover, EAE-PR-treated cells had a higher number of viable HEK-293 cells (a smaller number of cells with a red fluorescence) compared with the H_2_O_2_, EAE-PL, and AA treatment groups (Figure 3c). Moreover, the DCFH-DA staining results indicate a high ROS level in the H_2_O_2_ treatment group compared with control cells. A significant reduction in ROS status was observed in EAE-PR-treated cells compared with other samples (EAE-PL and AA), indicating a reduction in ROS-mediated toxicity (Figure 3d). However, a previous study also reported similar findings, where organic extracts of *Xylaria nigripes* significantly protected PC12 (rat pheochromocytoma) cells from H_2_O_2_-mediated cell damage by preventing lactate dehydrogenase release, reducing DNA damage, restoring the mitochondrial membrane potential, and preventing aberrant apoptosis by upregulating Bcl-2 and downregulating Bax and caspase 3 [45].

### 3.7. Metabolite Profiling by UHPLC-Q-TOF-MS/MS

The UHPLC-Q-TOF-MS/MS technique is regarded as one of the most sensitive strategies for the accurate detection and quantification of a wide range of polar compounds in secondary metabolites, including phenolics and terpenoids. Hence, the potential compounds present in EAE-PL and EAE-PR were analyzed using UHPLC-Q-TOF-MS/MS and tentatively identified (Table 5 and Table 6). A total of 20 compounds were tentatively identified in the EAE-PL sample group as shown in Table 5. The EAE-PR sample group exhibited the presence of a total of 26 tentatively identified compounds (Table 6). However, sorbic acid, phenylacetic acid, and ferulic acid were identified in both the EAE-PL and EAE-PR sample groups. A recent LC-MS characterization documented the presence of sorbic acid in the EAE of endophytic *Chaetomium subaffine* and described it as a potent antibacterial compound [46]. Phenylacetic acid and 3-indoleacetic acid, classified as plant hormones (auxins), could be produced by fungi as antimicrobial and antioxidant agents to assist with biological control and help plant growth [47,48]. In some recent studies, ferulic acid was reported to be one of the major phenolics in the extract of endophytic *P. roqueforti* and *P. oxalicum* [20,49]. In addition, vanillin, scopoletin, and sinapic acid were identified as major constituents of EAE-PR (Table 6). Moreover, the antioxidant and cytotoxic EAE of *P. citrinum* exhibited the presence of vanillin and coumaric acid in a UHPLC analysis [50]. It has been reported that the biosynthesis of secondary metabolites, such as scopoletin and isofraxidin, occurs via the shikimic acid pathway within the endophytic system [51]. However, Brader et al. reported that isofraxidin is also involved in the biosynthesis of scopoletin, aesculetin, and fraxetin, which were also identified in the EAE-PR (Table 6) [52]. In addition, sinapic acid has been identified to be a major phenolic acid in the EAE of marine-derived *P. brevicompactum* and is reported to have antioxidant and cytotoxic activities [53]. Similarly, maculosin has been reported to be an antimicrobial agent and is isolated from the EAE of endophytic *P. chrysogenum* [54]. Overall, the UHPLC-Q-TOF-MS/MS findings indicate a higher abundance of compounds in the EAE-PR sample group compared with the EAE-PL samples (Table 5 and Table 6). Perhaps this could be the reason for the higher bioactivities (tested in this study) of the EAE-PR compared with the EAE-PL.

Statistical measures (Unsupervised) such as PCA are utilized to minimize the number of variables in multivariate data while keeping the majority of the variation [55]. The significant differences between the EAE-PL and EAE-PR sample datasets (three groups from each sample set) were confirmed by multivariate statistical analysis (Figure 4). Therefore, the PCA analysis was utilized to examine the discriminative metabolites and identify metabolic differences between the samples (EAE-PL and EAE-PR). The PCA score plot indicated two sample clusters (EAE-PL and EAE-PR), reflecting the differentiation between the two groups. The first two principal components (PCs) exhibited 100% of the total variance (PC1, 100%; PC2, 0%), indicating the clear separation of both the samples and the metabolites and suggesting a high degree of diversity among the EAE-PL and EAE-PR samples (Figure 4a). Furthermore, the statistical variation in the dataset of the sample groups was established by the heatmap generated using the cluster separation results. The heatmap demonstrated the qualitative composition and the relative abundance of the compounds tentatively identified in the EAE-PL and EAE-PR (Figure 4b). Three groups in the heatmap indicate the replicate value of each sample set (EAE-PL and EAE-PR). The analyzed compounds can be clustered using a heatmap analysis based on concentrations, where the color pattern from red to blue (in this study) depicts concentrations in descending order [29].

The KEGG database was used to analyze the dominant pathways correlated with the tentatively identified metabolites (EAE-PL and EAE-PR) and the metabolism of differentially regulated metabolites as presented in Figure 5. However, quite different dominant pathways were observed in P. lanosum and P. radiatolobatum-associated metabolites (Figure 5a,b). The results indicate that all metabolites (tentatively identified in the EAE-PL and EAE-PR) followed three major pathways, including biosynthesis of secondary metabolites, phenylalanine metabolism, and biosynthesis of phenylpropanoids. It is evident that these pathways are positively correlated with the generation of phenolic acids (ferulic acid, p-coumaric acid, and sinapic acid) from a phenylalanine precursor [56]. Moreover, the pathway associated with the biosynthesis of plant hormones (Figure 5b) could be positively correlated with the synthesis of phenylacetic acid and 3-indoleacetic acid (Table 6).

### 3.8. Molecular Docking Analysis

Along with biological properties, the investigation of the physicochemical properties of candidate compounds is crucial for the development of therapeutic agents. So, the tentatively identified compounds of EAE-PL and EAE-PR were examined in order to predict their pharmacokinetic, physicochemical, drug-likeness, and related parameters according to Lipinski’s rule of five [10]. However, the compounds identified in the EAE-PL and EAE-PR did not violate any of Lipinski’s five rules (Appendix A). In this study, since the EAE-PR demonstrated substantially higher in vitro bioactivities than the EAE-PL, the compounds tentatively identified in the EAE-PR were selected for in silico molecular docking studies. In molecular docking experiments, the hypothesized antidiabetic, antioxidant, and antibacterial activities of mycocompounds and their probable mode of action were deduced on the respective target proteins (α-amylase and α-glucosidase, NADPH-oxidase, and D-alanine D-alanine ligase) via their molecular interaction at the atomic level. α-amylase and α-glucosidase inhibitory compounds from natural sources have shown potential responses in hyperglycemia management and have attracted researchers from around the world. The NADPH oxidases utilize the Nox catalytic subunit to catalyze the transfer of electrons from NADPH to molecular O_2_ to generate ROS such as superoxide or H_2_O_2_. In bacteria, D-alanine D-alanine ligases are essential ATP-dependent enzymes participating in cell wall (peptidoglycan) biosynthesis. Therefore, these enzymes (NADPH oxidase and D-alanine D-alanine ligase) are considered prominent targets for antioxidant and antibiotic development.

The molecular docking study was precisely focused on the free binding energy, H-bonding, carbon–hydrogen (C-H) bonding, and Van der Waals interactions (VDIs). The H-bonding and VDIs are associated with the binding interactions, whereas the C-H bonding and Pi–sigma interactions are associated with the stability of ligands (selected compounds) and the receptor-docked complex [57]. The scores of docking binding energies among the binding sites of target (α-amylase and α-glucosidase, NADPH-oxidase, and D-alanine D-alanine ligase) proteins are presented in Table 7. In addition, the H-bonding, C-H bonding, and VDIs with amino acids involved at the binding sites of α-amylase and α-glucosidase (Appendix A) and NADPH oxidase and D-alanine D-alanine ligase (Appendix A) against the ligands during the docking run are also summarized. Among the selected ligands, thiophene A exhibited the highest binding energy (kcal/mol) with the target α-amylase (−9.75), α-glucosidase (−11.74), NADPH oxidase (−10.96), and D-alanine D-alanine ligase (−12.06) proteins (Table 7). The docking simulation of thiophene A with the crystal structure of α-amylase and α-glucosidase during the *in silico* molecular interaction is shown in Figure 6a–d. Interaction of thiophene A with α-amylase was not related to H-bonding and C-H bonding, but was associated with 11 VDIs (TRP58, TRP59, GLU60, GLN63, GLY104, SER105, GLY106, GLY164, ARG195, ASP197, and ASP300) (Figure 6b and Appendix A). With α-glucosidase, thiophene A interacted through three VDIs (THR700, ALA704, and GLN776) and other non-covalent interactions (Figure 6d and Appendix A). Furthermore, Figure 7a–d depict the in silico molecular docking simulation of thiophene A with the crystal structure of NADPH oxidase and D-alanine D-alanine ligase. However, the docking of thiophene A with NADPH oxidase and D-alanine D-alanine ligase indicated six VDIs (HIS10, PHE14, GLY59, ARG308, PRO816, and ARG882) and five VDIs (ASN129, GLU203, GLU451, ASP590, and SER589), respectively (Figure 7b,d and Appendix A).

The docking simulation of hematommic acid with α-amylase and limonene with α-glucosidase can be seen in Appendix A–d. Moreover, the docking simulation of limonene with NADPH oxidase and D-alanine D-alanine ligase is shown in Appendix A–d. Similarly, previous studies also predicted the enzyme inhibitory activity (α-glucosidase inhibition) of the compounds present in the EAE of endophytic *Diaporthe* sp. and the bioactivities (antioxidant and antibacterial) of the compounds produced by endophytic *P. pinophilum* using molecular docking studies [9,22]. The molecular docking observations (made in this study) suggested that compounds with noteworthy α-amylase and α-glucosidase inhibition, antioxidant, and antibacterial potential are present in the EAE-PR.

## 4. Conclusions

Novel therapeutic agents derived from natural sources could help us to address persistent/future concerns related to NCDs, bacterial infections, and antibacterial resistance. This study examined and compared the antidiabetic, antioxidant, antibacterial, and cytotoxic activities of the ethyl acetate extracts of two endophytic *Penicillium* sp. (EAE-PL and EAE-PR). Compared with EAE-PL, the EAE-PR showed higher α-amylase and α-glucosidase inhibition, free radical scavenging, antibacterial activity, and protective effects against H_2_O_2_ in HEK-293 cells. The UHPLC-Q-TOF-MS/MS studies manifested the presence of bioactive compounds in the EAE-PR. Furthermore, in silico molecular docking interactions of bioactive compounds, namely thiophene A, limonene, and phenylacetic acid, against α-amylase, α-glucosidase, NADPH oxidase, and D-alanine D-alanine ligase indicate the possible reason for the antidiabetic, antioxidant, and antibacterial activities of the EAE-PR. Yet, further investigations are needed to fractionate and characterize specific bioactive compounds for the development, validation, and design of potent therapeutic drugs.

## Figures and Tables

**Figure 1 antioxidants-12-00248-f001:**
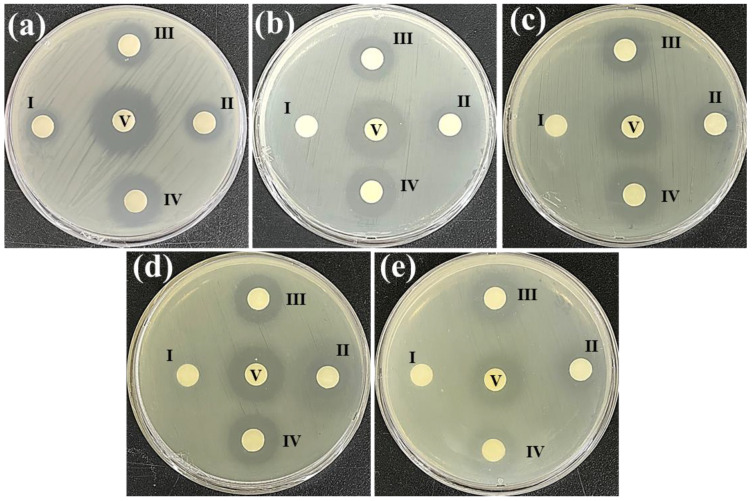
Antibacterial activity of the EAE-PL and EAE-PR against *B. cereus* (**a**), *S. aureus* (**b**), *L. monocytogenes* (**c**), *E. coli* (**d**), and *S. enterica* (**e**) in the disc diffusion assay. I (500 µg/mL) and II (1000 µg/mL) are the EAE-PL treatments. III (500 µg/mL) and IV (1000 µg/mL) are the EAE-PR treatments. V (100 µg/mL) is the experimental control (erythromycin (Ery)).

**Figure 2 antioxidants-12-00248-f002:**
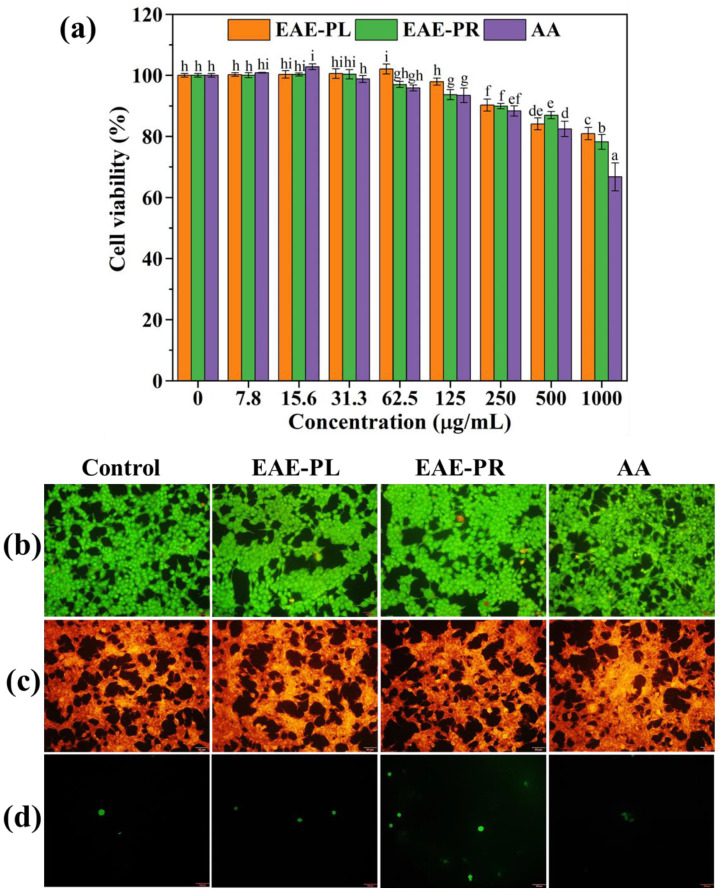
Cell viability analysis of HEK-293 cells against the EAE-PL and EAE-PR using an MTT assay (**a**) and a florescent staining assay (**b**–**d**), where the AO/EB dual staining (**b**), Rh-123 (**c**), and DCFH-DA (**d**) staining examinations were performed with a 50 µm scale bar. Error bars represent the SD of 3 independent experiments, and superscripts on error bars represent significance (*p* < 0.05) among the samples.

**Figure 3 antioxidants-12-00248-f003:**
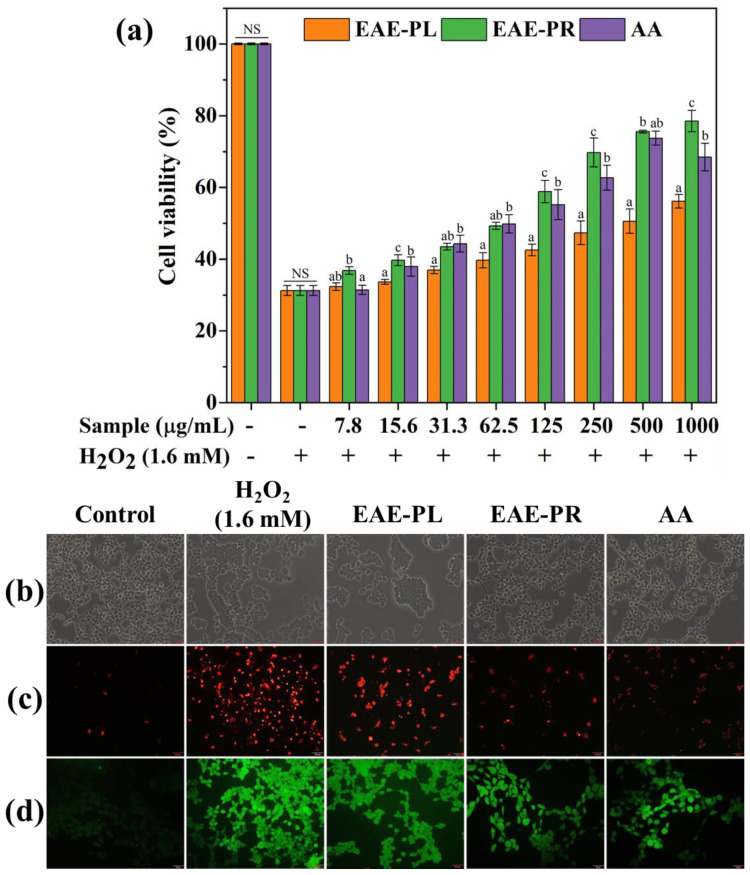
Analysis of the cytoprotective effect of the EAE-PL and EAE-PR in HEK-293 cells against H_2_O_2_ stress using an MTT assay (**a**), light microscopy (**b**), and a florescent staining assay (**c**,**d**), where the light microscopy (**b**), PI (**c**), and DCFH-DA (**d**) staining examinations were performed with a 50 µm scale bar. Error bars represent the SD of 3 independent experiments and superscripts on error bars represent significance (*p* < 0.05) among the groups.

**Figure 4 antioxidants-12-00248-f004:**
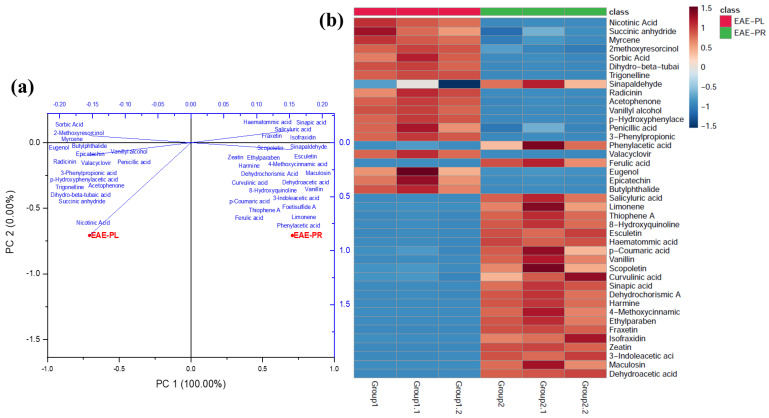
The principal component analysis (PCA) of the EAE-PL and EAE-PR by comparing PC1 with PC2 (**a**) and the heatmap of the metabolites (levels) tentatively identified in the EAE-PL and EAE-PR samples from red to blue in descending order (**b**).

**Figure 5 antioxidants-12-00248-f005:**
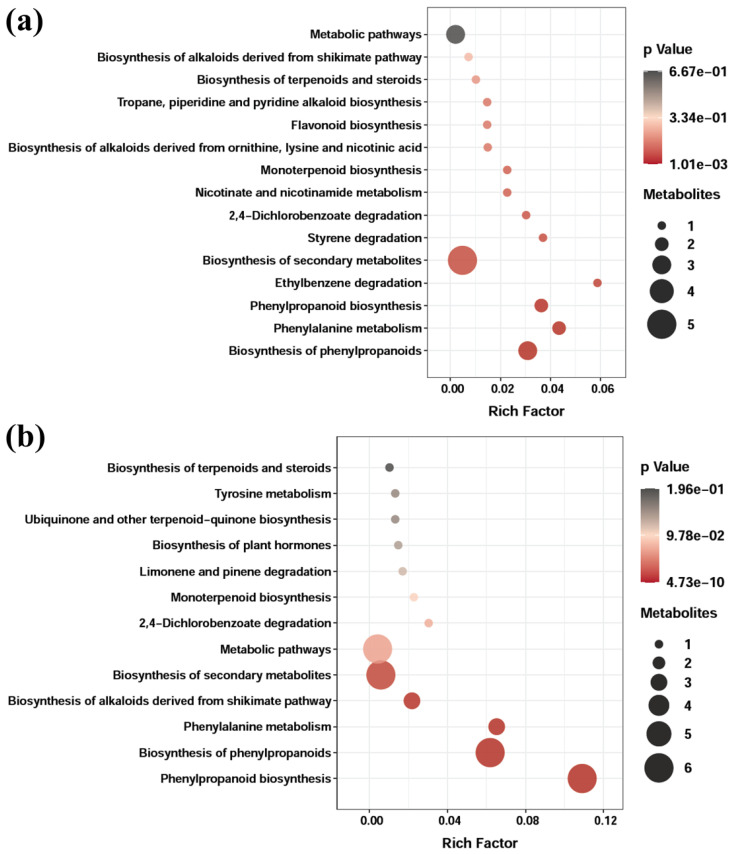
Scatter plot of the KEGG pathway analysis of differentially regulated metabolites present in EAE-PL (**a**) and EAE-PR (**b**). The X-axis denotes the Rich Factor (the ratio of the number of differentially regulated metabolites annotated in a pathway indicated on the Y-axis), and the Y-axis denotes the name of the pathways. The color of the bubbles represents the significance of the regulation at the magnitude of the *p*-value.

**Figure 6 antioxidants-12-00248-f006:**
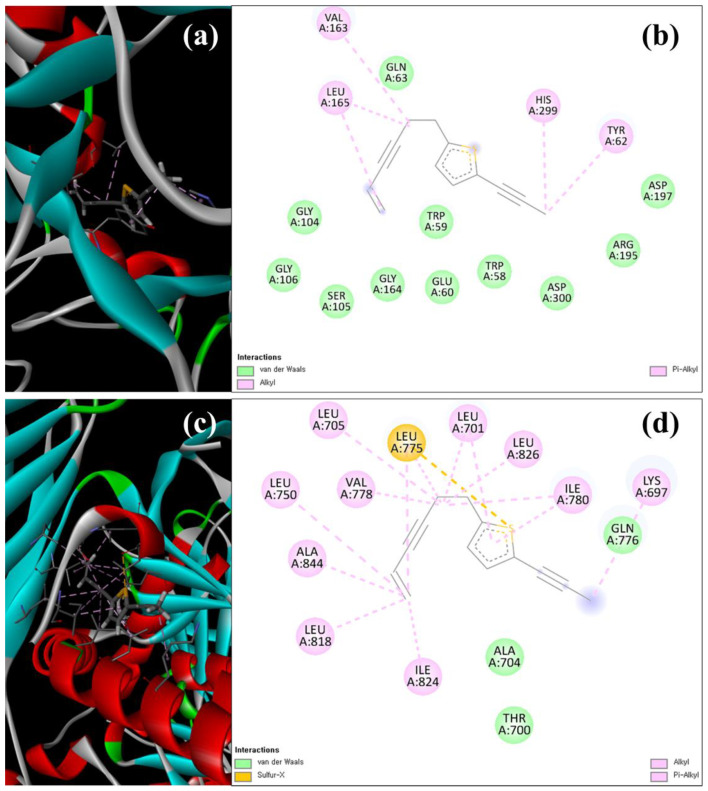
The structural overlay and 3D (**a**,**c**), and 2D (**b**,**d**) docking simulation of thiophene A with the crystal structure of α-amylase (**a**,**b**) and α-glucosidase (**c**,**d**) during in silico molecular interaction analysis. The 3D simulations represent target proteins (thick tubes), ligands (thin tubes), and interaction forces/bonds (dotted tubes).

**Figure 7 antioxidants-12-00248-f007:**
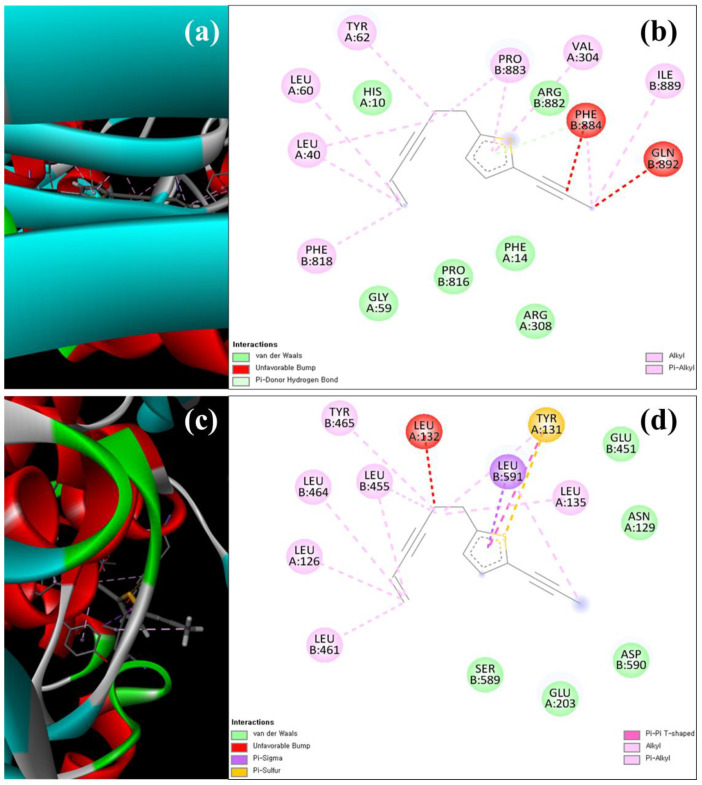
The structural overlay and 3D (**a**,**c**), and 2D (**b**,**d**) docking simulation of thiophene A with the crystal structure of NADPH oxidase (**a**,**b**) and D-alanine D-alanine ligase (**c**,**d**) during in silico molecular interaction analysis. The 3D simulations represent target proteins (thick tubes), ligands (thin tubes), and interaction forces/bonds (dotted tubes).

**Table 1 antioxidants-12-00248-t001:** Analysis of the total yield, total phenolic content, and total flavonoid content of the EAE-PL and EAE-PR. The superscripts represent significance (*p* < 0.05) among samples.

Sample	Extract Yield (%)	Total Phenolic Content (mg of GAE/g DW)	Total Flavonoid Content (mg of QE/g DW)
**EAE-PL**	0.41 ± 0.08 ^a^	17.89 ± 1.63 ^a^	3.31 ± 0.96 ^a^
**EAE-PR**	0.63 ± 0.05 ^b^	119.87 ± 3.74 ^b^	16.26 ± 1.95 ^b^

**Table 2 antioxidants-12-00248-t002:** Antioxidant activities and diabetes-related enzyme inhibitory activities of the EAE-PL and EAE-PR compared to ascorbic acid (AA) and acarbose (AC), respectively. ND, not determined. The superscripts represent significance (*p* < 0.05) among samples.

Sample	Half-Maximal Inhibitory Concentration (IC_50_; µg/mL)					
	α-amylase Inhibition	α-glucosidase Inhibition	ABTS^+^ Radical Scavenging	DPPH Radical Scavenging	Ferric-Reducing Power	Peroxyl Radical Scavenging
**EAE-PL**	>1000 ^c^	>1000 ^c^	59.5 ± 1.54 ^c^	>1000 ^c^	733.3 ± 2.61 ^c^	>1000 ^c^
**EAE-PR**	362.5 ± 2.50 ^b^	525 ± 1.38 ^b^	37.5 ± 3.46 ^a^	187.5 ± 3.92 ^a^	175.0 ± 1.46 ^a^	532.36 ± 2.52 ^b^
**AA**	ND	ND	49.25 ± 2.10 ^b^	241.6 ± 1.95 ^b^	326.5 ± 3.78 ^b^	521.86 ± 1.75 ^a^
**AC**	37.5 ± 2.3 ^a^	87.5 ± 2.58 ^a^	ND	ND	ND	ND

**Table 3 antioxidants-12-00248-t003:** Summary of the antibacterial activity of the EAE-PL and EAE-PR against Gram-positive and Gram-negative pathogens compared to the standard antibiotic erythromycin (Ery) in the disc diffusion assay. The superscripts followed by the value represent significance (*p* < 0.05) among the groups.

Bacterial Strains	Zone of Inhibition (mm)				
	EAE-PL (µg/mL)		EAE-PR (µg/mL)		Ery (100 µg/mL)
	500	1000	500	1000	
** *B. cereus* **	9 ± 0.5 ^b^	11.5 ± 1.0 ^d^	14 ± 0.3 ^c^	19 ± 1.5 ^b^	22 ± 1.5 ^b^
** *S. aureus* **	0	8 ± 0.3 ^a^	11 ± 0.1 ^a^	18 ± 0.5 ^a^	22 ± 0.3 ^b^
** *L. monocytogenes* **	0	9 ± 0.2 ^b^	15 ± 0.5 ^d^	19 ± 1.5 ^b^	21 ± 1.0 ^a^
** *E. coli* **	6.5 ± 0.2 ^a^	11.5 ± 0.5 ^e^	17 ± 0.1 ^e^	19 ± 1.0 ^b^	22 ± 0.5 ^b^
** *S. enterica* **	0	10 ± 0.3 ^c^	13 ± 0.2 ^b^	18 ± 0.5 ^a^	21 ± 0.8 ^a^

**Table 4 antioxidants-12-00248-t004:** Determination of the minimum inhibitory concentration (MIC) and minimum bactericidal concentration (MBC) of the EAE-PL and EAE-PR against Gram-positive and Gram-negative pathogens using a microtiter assay. The superscripts followed by the value represent significance (*p* < 0.05) among samples.

Bacterial Strains	MIC (µg/mL)		MBC (µg/mL)	
	EAE-PL	EAE-PR	EAE-PL	EAE-PR
** *B. cereus* **	62.5 ^a^	31.25 ^a^	500 ^a^	125 ^a^
** *S. aureus* **	500 ^d^	125 ^c^	1000 ^b^	500 ^c^
** *L. monocytogenes* **	500 ^d^	31.25 ^a^	1000 ^b^	125 ^a^
** *E. coli* **	125 ^b^	62.5 ^b^	500 ^a^	250 ^b^
** *S. enterica* **	250 ^c^	62.5 ^b^	1000 ^b^	250 ^b^

**Table 5 antioxidants-12-00248-t005:** Metabolite profiling of the ethyl acetate extract of *Penicillium lanosum* (EAE-PL) by LC-MS/MS analysis. RT, retention time.

S.No.	RT (min)	Tentatively Identified Compounds	Formula	Molecular Weight	Found at Mass	Area	Adduct/Charge
1.	1.23	Nicotinic Acid	C_6_H_5_NO_2_	123.03215	124.0388	2.2 × 10^6^	[M+H]+
2.	1.64	Succinic anhydride	C_4_H_4_O_3_	100.01648	101.0233	2.4 × 10^5^	[M+H]+
3.	2.03	Myrcene	C_10_H_16_	136.12567	137.1324	3.0 × 10^4^	[M+H]+
4.	3.33	2-Methoxyresorcinol	C_7_H_8_O_3_	140.04776	141.0544	3.4 × 10^6^	[M+H]+
5.	5.17	Sorbic Acid	C_6_H_8_O_2_	112.05289	113.0597	1.1 × 10^6^	[M+H]+
6.	6.42	Dihydro-beta-tubaic acid	C_13_H_16_O_4_	236.10549	237.1124	4.6 × 10^5^	[M+H]+
7.	7.35	Trigonelline	C_7_H_7_NO_2_	137.04816	138.055	5.7 × 10^5^	[M+H]+
8.	9.12	Sinapaldehyde	C_11_H_12_O_4_	208.07402	209.0809	4.5 × 10^5^	[M+H]+
9.	9.15	Radicinin	C_12_H_12_O_5_	236.06843	237.0751	1.1 × 10^6^	[M+H]+
10.	9.36	Acetophenone	C_8_H_8_O	120.05784	121.0645	4.3 × 10^6^	[M+H]+
11.	10.91	Vanillyl alcohol	C_8_H_10_O_3_	154.06361	155.0703	3.9 × 10^6^	[M+H]+
12.	11.41	p-Hydroxyphenylacetic acid	C_8_H_8_O_3_	152.04762	153.0543	2.8 × 10^7^	[M+H]+
13.	11.51	Penicillic acid	C_8_H_10_O_4_	170.05863	169.051	1.7 × 10^5^	[M-H]-
14.	13.08	3-Phenylpropionic acid	C_9_H_10_O_2_	150.06885	151.0755	7.1 × 10^5^	[M+H]+
15.	13.36	Phenylacetic acid	C_8_H_8_O_2_	136.05302	137.0598	5.3 × 10^5^	[M+H]+
16.	14.47	Valacyclovir	C_13_H_20_N_6_O_4_	324.15505	325.1615	7.2 × 10^6^	[M+H]+
17.	15.85	Ferulic acid	C_10_H_10_O_4_	194.05872	195.0655	1.2 × 10^6^	[M+H]+
18.	16.54	Eugenol	C_10_H_12_O_2_	164.08434	165.091	1.0 × 10^6^	[M+H]+
19.	16.61	Epicatechin	C_15_H_14_O_6_	290.07957	291.0866	1.2 × 10^6^	[M+H]+
20.	16.77	Butylphthalide	C_12_H_14_O_2_	190.10013	191.1068	1.7 × 10^6^	[M+H]+

**Table 6 antioxidants-12-00248-t006:** Metabolite profiling of the ethyl acetate extract of *Penicillium radiatolobatum* (EAE-PR) by LC-MS/MS analysis. RT, retention time.

S.No.	RT (min)	Tentatively Identified Compounds	Formula	Molecular Weight	Found at Mass	Area	Adduct/Charge
1.	0.88	Salicyluric acid	C_9_H_9_NO_4_	195.05402	196.0607	1.9 × 10^5^	[M+H]+
2.	2.4	Limonene	C_10_H_16_	136.12562	137.1325	1.1 × 10^5^	[M+H]+
3.	2.59	Thiophene A	C_13_H_8_S	196.03571	197.0424	4.8 × 10^5^	[M+H]+
4.	3.44	Zeatin	C_10_H_13_N_5_O	219.11283	220.1194	9.0 × 10^5^	[M+H]+
5.	5.17	Sorbic Acid	C_6_H_8_O_2_	112.05289	113.0597	2.5 × 10^5^	[M+H]+
6.	6.82	3-Indoleacetic acid	C_10_H_9_NO_2_	175.06414	176.0708	6.2 × 10^5^	[M-H]-
7.	7.93	8-Hydroxyquinoline	C_9_H_7_NO	145.05328	146.06	4.7 × 10^5^	[M+H]+
8.	8.13	Maculosin	C_14_H_16_N_2_O_3_	260.11727	261.1238	1.1 × 10^6^	[M+H]+
9.	8.58	Dehydroacetic acid	C_8_H_8_O_4_	168.04297	169.0497	9.0 × 10^5^	[M+H]+
10.	8.68	Esculetin	C_9_H_6_O_4_	178.02731	179.0342	3.4 × 10^5^	[M+H]+
11.	9.12	Sinapaldehyde	C_11_H_12_O_4_	208.07402	209.081	4.9 × 10^5^	[M+H]+
12.	9.81	Foetisulfide A	C_8_H_16_OS_3_	224.03749	225.0438	5.8 × 10^3^	[M+H]+
13.	10.93	Haematommic acid	C_9_H_8_O_5_	196.03785	197.0447	3.8 × 10^5^	[M+H]+
14.	11.08	p-Coumaric acid	C_9_H_8_O_3_	164.04811	165.0547	1.4 × 10^6^	[M+H]+
15.	12.28	Vanillin	C_8_H_8_O_3_	152.04761	153.0542	2.6 × 10^7^	[M+H]+
16.	12.73	Scopoletin	C_10_H_8_O_4_	192.04292	193.0495	1.3 × 10^7^	[M+H]+
17.	12.76	Curvulinic acid	C_10_H_10_O_5_	210.05359	209.046	1.3 × 10^6^	[M-H]-
18.	12.89	Sinapic acid	C_11_H_12_O_5_	224.06794	225.0753	1.2 × 10^7^	[M+H]+
19.	14.6	Dehydrochorismic Acid	C_10_H_8_O_6_	224.03286	225.0395	4.5 × 10^5^	[M+H]+
20.	14.98	Harmine	C_13_H_12_N_2_O	212.09586	213.1027	3.4 × 10^5^	[M+H]+
21.	15.01	4-Methoxycinnamic acid	C_10_H_10_O_3_	178.06369	179.0704	1.3 × 10^6^	[M+H]+
22.	15.84	Ethylparaben	C_9_H_10_O_3_	166.06370	167.0705	1.9 × 10^6^	[M+H]+
23.	15.85	Ferulic acid	C_10_H_10_O_4_	194.05872	195.0646	2.6 × 10^7^	[M+H]+
24.	16.05	Fraxetin	C_10_H_8_O_5_	208.03726	209.0439	8.7 × 10^6^	[M+H]+
25.	16.41	Isofraxidin	C_11_H_10_O_5_	222.05344	221.0459	4.0 × 10^5^	[M-H]-
26.	16.54	Phenylacetic acid	C_8_H_8_O_2_	136.05302	137.0598	1.1 × 10^6^	[M+H]+

**Table 7 antioxidants-12-00248-t007:** Free binding energies of the tentatively identified compounds from the EAE-PR within the active pockets of α-amylase (PDB: 1OSE), α-glucosidase (PDB: 5NN8), NADPH oxidase (PDB: 2CDU), and D-alanine D-alanine ligase (PDB: 2PVP) during the in silico molecular docking analysis. ND, not determined.

Compounds	α-amylase	α-glucosidase	NADPH Oxidase	D-alanine D-alanine Ligase
3-Indoleacetic acid	−8.77	−9.60	−8.88	−9.11
4-Methoxycinnamic acid	−8.28	−8.06	−8.80	−9.17
8-Hydroxyquinoline	−8.35	−8.85	−8.47	−9.04
Curvulinic acid	−8.07	−8.63	−8.59	−9.04
Dehydroacetic acid	−6.54	−6.99	−7.33	−6.87
Dehydrochorismic Acid	−7.59	−8.39	−8.05	−8.51
Esculetin	−8.22	−8.14	−8.15	−8.39
Ethylparaben	−8.50	−8.43	−8.33	−8.79
Ferulic acid	−7.85	−8.65	−8.53	−8.93
Foetisulfide A	−8.31	−9.73	−8.99	−10.39
Fraxetin	−7.95	−7.70	−7.80	−8.13
Haematommic acid	−9.00	−8.52	−7.86	−8.36
Harmine	−8.14	−8.06	−8.27	−8.08
Isofraxidin	−7.99	−8.03	−8.10	−7.72
Limonene	−8.78	−10.17	−9.65	−12.40
Maculosin	−8.75	−8.83	−9.28	−8.75
p-Coumaric acid	−8.51	−9.75	−9.06	−9.25
Phenylacetic acid	−8.75	−9.99	−8.87	−10.61
Salicyluric acid	−8.34	−8.63	−8.70	−9.26
Scopoletin	−7.60	−7.28	−8.27	−8.05
Sinapaldehyde	−7.87	−8.39	−8.80	−8.60
Sinapic acid	−8.54	−7.84	−8.59	−8.45
Sorbic Acid	−7.99	−9.57	−7.73	−9.09
Thiophene A	−9.75	−11.74	−10.96	−12.06
Vanillin	−7.29	−8.09	−7.81	−8.12
Zeatin	−7.85	−7.91	−7.64	−7.87
Acarbose	−6.81	−7.52	ND	ND
Ascorbic acid	ND	ND	−7.47	ND
Erythronolide A (Ery)	ND	ND	ND	−9.07

## Data Availability

The data presented in this article are available on request.

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
