# Peer review of "Comparative Analysis of the Antioxidant, Antidiabetic, Antibacterial, Cytoprotective Potential and Metabolite Profile of Two Endophytic Penicillium spp."

_antioxidants, 2023, doi:10.3390/antiox12020248_

Round 1

Reviewer 1 Report

antioxidants-2143177

Comparative analysis of antioxidant, antidiabetic, antibacterial and cytoprotective potential, and metabolite profile of two endophytic Penicillium sp. by experimental and bioinformatics approach

Kumar Vishven Naveen , Kandasamy Saravanakumar , Anbazhagan Sathiyaseelan , Myeong-Hyeon Wang

In this investigation authors analyzed the potential properties of endophytic Penicillum by using experimental and bioinformatics approaches. They observed that metabolites, present in ethyl acetate extracts from Penicillum lanosum and Penicillum radiatolobatum, exert several properties including antioxidant, cytoprotective, antibacterial and antidiabetic effects, making them noteworthy to further investigations in therapeutic strategies.

In this work, authors wanted to explore novel compounds as a new strategy to find out natural therapeutic agents with minor side effects. This is an interesting issue; the experimental study is well designed, and the paper is well written; however, the manuscript is incomplete.

I could not find supplementary materials. Therefore, several figures are missing as well as the description of some assays in materials and methods section, and consequently part of the discussion and of the conclusions is not supported by the results.

Author Response

  1. In this investigation authors analyzed the potential properties of endophytic Penicillum by using experimental and bioinformatics approaches. They observed that metabolites, present in ethyl acetate extracts from Penicillum lanosum and Penicillum radiatolobatum, exert several properties including antioxidant, cytoprotective, antibacterial and antidiabetic effects, making them noteworthy to further investigations in therapeutic strategies. In this work, authors wanted to explore novel compounds as a new strategy to find out natural therapeutic agents with minor side effects. This is an interesting issue; the experimental study is well designed, and the paper is well written; however, the manuscript is incomplete.

Response: Thank you very much for your encouraging words and valuable comments for the betterment of our manuscript. We have revised the manuscript considering the reviewer’s suggestion and the revision is highlighted in red color.

  1. I could not find supplementary materials. Therefore, several figures are missing as well as the description of some assays in the materials and methods section, and consequently part of the discussion and of the conclusions is not supported by the results.

Response: This technical error is deeply regretted. Thank you very much for raising genuine concerns for the betterment of our manuscript. However, we have rectified the concerned issue in the revised manuscript and supplementary materials file.

Reviewer 2 Report

The manuscript deals with two endophytic Penicillium spp. – chemical composition of ethyl acetate extracts and their alpha (α)-amylase and α-glucosidase inhibitory, antioxidant, and antibacterial activity. The paper is well written and organized, and deserves publication. However, several points should be clarified/indicated.

1. Tables 2 and 4 –How many repetitions have been performed? Please indicate standard deviation.

2. Please, indicate what is behind the six groups in PCA and heatmap.

3. Are the both extracts analyzed at the same chromatographic conditions? The same compounds but at different RT have been detected in both extracts.

Author Response

The manuscript deals with two endophytic Penicillium spp. – chemical composition of ethyl acetate extracts and their alpha (α)-amylase and α-glucosidase inhibitory, antioxidant, and antibacterial activity. The paper is well written and organized, and deserves publication. However, several points should be clarified/indicated.

Response: Thank you very much for your encouraging words and valuable comments for the betterment of our manuscript. We have revised the manuscript as per your suggestion and the revision is highlighted in red color.

  1. Tables 2 and 4 –How many repetitions have been performed? Please indicate standard deviation.

Response: We truly appreciate your keen observation regarding the betterment of our manuscript. The experiments are performed in three independent repetitions. As per your suggestion, the SD and statistical significance are incorporated in the respective tables in the revised manuscript.

  1. Please, indicate what is behind the six groups in PCA and heatmap.

Response: Thank you very much for your valuable suggestion. A total of six groups (Three groups from each sample set) in the heatmap indicates the replicate value of each sample set (EAE-PL and EAE-PR). However, considering your suggestion, the confusing statement is corrected and a descriptive PCA plot and its description are incorporated for a better representation of the results in the revised manuscript.

  1. Are both the extracts analyzed at the same chromatographic conditions? The same compounds but at different RT have been detected in both extracts.

Response: We are very thankful for your keen observation and for pointing out the confusing glitch. Yes! both extracts were analyzed at the same chromatographic conditions. We think the RT was mentioned incorrectly due to typographical or formatting errors. We have re-checked the whole table and corrected it as per your kind remark and we have also incorporated the adduct or charge of the individual compounds.

Reviewer 3 Report

In order to be considered for publication, the manuscript presented for review must undergo a major revision.

The main points to be addressed are:

1. The introduction should focus more on the literature data regarding composition and action of Penicillium extracts

2. A negative control should be used for the antimicrobial assay

3. The discussion part should be greatly enhanced. What are the presumed correlation composition/effect? What are the differences between this study and literature data? The results should be compared and discussed considering literature data

4. The Supplementary material should be provided for the revised manuscript 

Minor issues:

1. Provide city/country for reagent and equipment producers, according MDPI template

2. Minor typos and language errors should be corrected in the manuscript (see r. 195, 285-286, etc)

3. Table 2 should also present the statistical significance of the differences for the obtained results   

Author Response

In order to be considered for publication, the manuscript presented for review must undergo a major revision.

Response: Thank you very much for reviewing our manuscript and for your valuable comments for the betterment of our manuscript. We have revised the manuscript as per your suggestion and the revision is highlighted in red color.

The main points to be addressed are:

1. The introduction should focus more on the literature data regarding composition and action of Penicillium extracts.

Response: We agree with you and thank you for the valuable comments. As you suggested the revision is made in the ‘Introduction’ section of the revised manuscript and highlighted in red color.

2. A negative control should be used for the antimicrobial assay.

Response: Thank you! As the reviewer knows, antimicrobial assays generally involve PBS as a negative control. Since we dissolved our samples in the PBS, we have also used PBS as the medium throughout our antimicrobial assays. The concerned writing glitch is deeply regretted. However, as per your suggestion, this information is stated in the revised manuscript.

3. The discussion part should be greatly enhanced. What are the presumed correlation composition/effect? What are the differences between this study and literature data? The results should be compared and discussed considering literature data.

Response: We truly appreciate your keen observation and for raising these concerns for the betterment of our manuscript. The ‘discussion’ section is revised as per your valuable suggestion and highlighted in red color.

4. The Supplementary material should be provided for the revised manuscript

Response: This technical error is deeply regretted. Thank you very much for raising genuine concerns for the betterment of our manuscript. However, we have rectified the concerned issue in the revised manuscript and supplementary materials file.

Minor issues:

1. Provide city/country for reagent and equipment producers, according MDPI template.

Response: Thank you very much for your keen observation and valuable suggestion for the betterment of our manuscript. The same is rectified in the revised manuscript.

2. Minor typos and language errors should be corrected in the manuscript (see r. 195, 285-286, etc)

Response: Thank you for catching the formatting glitch. As you suggested the correction is made at the respective place in the revised manuscript and highlighted in red color.

3. Table 2 should also present the statistical significance of the differences for the obtained results.

Response: We truly appreciate your keen observation regarding the betterment of our manuscript. The experiments are performed in three independent repetitions. As per your suggestion, the SD and statistical significance are incorporated in the concerned table in the revised manuscript.

Round 2

Reviewer 1 Report

The authors addressed reviewer’s comments and added the missing materials.  The quality of the manuscript has been improved. I have no further questions currently.

Author Response

Thank you very much for reviewing our manuscript and for your valuable comments for the betterment of our manuscript.

Reviewer 3 Report

The authors managed to answer in a satisfactory manner to the reviewer's comments. I have only one observation: Abstract, row 17- what does "did not cause considerable toxicity in the HEK-293 cell line" stand for? Please use a more scientific description: i.e. did not exhibited significant toxicity compared with control, etc. 

Author Response

The authors managed to answer in a satisfactory manner to the reviewer's comments. I have only one observation: Abstract, row 17- what does "did not cause considerable toxicity in the HEK-293 cell line" stand for? Please use a more scientific description: i.e. did not exhibited significant toxicity compared with control, etc. 

Response: Thank you very much for reviewing our manuscript and for your valuable comments for the betterment of our manuscript. We have revised the manuscript as per your suggestion and the revision is highlighted in red color.

We believe that we have answered all the queries asked and made all the possible changes in the text as per reviewers’ and editors’ suggestions. We are looking forward to the valuable decision on our submission.

Thank you!

Sincerely,
